# The Investigation of Gender Differences in Subjective Wellbeing in Children and Adolescents: The UP&DOWN Study

**DOI:** 10.3390/ijerph17082732

**Published:** 2020-04-15

**Authors:** Sara Esteban-Gonzalo, Laura Esteban-Gonzalo, Verónica Cabanas-Sánchez, Marta Miret, Oscar L. Veiga

**Affiliations:** 1Faculty of Biomedicine, Psychology Department, Universidad Europea de Madrid, 28670 Madrid, Spain; 2Faculty of Biomedicine, Nursing Department, Universidad Europea de Madrid, 28670 Madrid, Spain; laura.esteban@uam.es; 3Faculty of Medicine, Nursing Department, Universidad Autónoma de Madrid, 28029 Madrid, Spain; 4CEI UAM + CSIC, IMDEA Food Institute, 28049 Madrid, Spain; veronica.cabanas84@gmail.com; 5Department of Psychiatry, Universidad Autónoma de Madrid, Madrid, 28029 Madrid, Spain; marta.miret@uam.es; 6Instituto de Salud Carlos III, Centro de Investigación Biomédica en Red de Salud Mental, CIBERSAM, 28029 Madrid, Spain; 7Department of Psychiatry, Hospital Universitario de La Princesa, Instituto de Investigación Sanitaria Princesa (IIS-Princesa), 28006 Madrid, Spain; 8Department of Physical Education, Sport and Human Movement. Faculty of Teacher Training and Education, Universidad Autónoma de Madrid, 28049 Madrid, Spain; oscar.veiga@uam.es

**Keywords:** evaluative wellbeing, hedonic wellbeing, eudemonic wellbeing, children, adolescents, gender

## Abstract

Objective: Based on a three-factor model of subjective wellbeing (evaluative, hedonic and eudemonic), the purpose of this study was to analyze gender differences in children and adolescents through three different subjective wellbeing indicators. Method: The sample comprised 1.407 children and adolescents from Cadiz and Madrid (Spain), in the framework of the UP&DOWN study. Life satisfaction was measured with the subjective happiness scale, positive and negative affect were measured with the positive and negative affect schedule, and purpose in life was assessed with the children’s hope scale. Results: Linear regression models indicate the existence of significant gender differences only in adolescents, with higher scores among girls in positive affect (*p* = 0.016) and negative affect (*p* < 0.001) but with lower scores in purpose in life (*p* = 0.024). Conclusions: These results highlight the role of gender as an important factor in explaining differences in subjective wellbeing. Additionally, results indicate that gender differences in subjective wellbeing are observed in adolescents, but not in children, suggesting that the gender gap in subjective wellbeing begins at the age of 12. Mental health practitioners should pay attention to these findings in order to implement screening methods and interventions focused on these needs.

## 1. Introduction

For many centuries, scientists have focused their efforts on analyzing the components of wellbeing. Theorists from different disciplines have suggested a variety of factors as important elements of wellbeing [1]. Since Ancient Greece with Aristoteles’ concept of happiness to now, wellbeing has been described from many perspectives [2].

Subjective wellbeing could be understood as one more approach among the multiple dimensions of wellbeing [3]. According to recent perspectives, three different components of subjective wellbeing should be distinguished: global life satisfaction (evaluative wellbeing), sense of meaning and purpose of life (eudemonic wellbeing) and levels of positive and negative affect that people experience in their everyday lives (hedonic wellbeing) [4,5,6,7,8,9].

Evaluative wellbeing is a cognitive component that refers to the level of satisfaction with life as a whole [10], and depends on a personal judgment process in which individuals evaluate their lives on the basis of their own criteria, assessing life situations and comparing these situations with those of others and in the past [11]. Life satisfaction is generally measured by asking respondents how satisfied they feel in their life, placing themselves in a given point on a given scale [10]. The eudemonic concept of wellbeing is focused on meaning and purpose in life and self-realization, and it defines wellbeing in terms of the degree to which people feel fulfilled as they function in different areas of life [12]. Eudemonic wellbeing emphasizes the importance of psychological constructs such as personal meaning and growth [8,13,14], which are usually measured through structured self-reported scales focused on multiple dimensions [14,15]. The hedonic wellbeing approach defines wellbeing in terms of pleasure and pain [12], considering feelings such as happiness, sadness, anger, stress and pain. It is commonly measured by analyzing positive and negative experiences in people’s daily lives with experience sampling methodologies (ESM) or similar methods based on diary techniques to appraise subjective experiences in daily life such as the day reconstruction method [6,16,17,18]. Empirical findings suggest that positive and negative affect should be separately measured as independent dimensions by asking people about their feelings at a given period of time [19].

### Gender and Wellbeing

During recent decades, biopsychosocial perspectives have emphasized the importance of sociocultural variables as predictors of wellbeing [20,21]. Links between gender and wellbeing have been established in order to understand inequalities in living conditions; gender-related factors may play an important role by contributing and reinforcing the impact of such differences [22,23].

Some studies have pointed out the existence of gender differences in the levels of happiness and affect [24,25]. Sociocognitive components of gender roles and traits may contribute to and enhance differences in stress, cognitive appraisal and coping strategies, supporting the impact of gender roles in such mechanisms [26]. Gender differences in fear and anxiety have been reported. A wide range of factors such as biological influences, temperamental factors, stress and trauma, cognitive factors and environmental factors could be behind such differences [27].

Studies on subjective wellbeing and gender in adolescents have offered explanations to justify gender differences in emotional states among adolescents and young adults [28,29,30,31,32,33,34]. For example, the gender intensification hypothesis [35] claims that in early adolescence, boys and girls experience higher pressure in terms of what is culturally accepted for each sex by parents, peers, educators and the media. To confront this pressure they adopt more differentiated gender-role identities, which are supposed to be adaptive for their future adult roles [29]. Although this hypothesis has not been demonstrated in contemporary adolescents, it has been found that increased masculinity and androgyny in both boys and girls are linked to psychological wellbeing [36,37]. In addition, similar studies suggest that masculinity may have positive effects on mood states because it is positively associated with self-efficacy, perceived competence [38] or self-esteem [39,40] in a context where social evaluation, internal and interpersonal distress, and body image-related concerns are specially salient for adolescent girls [31]. It has also been pointed out that gender differences in negative mood experiences begin around the ages of 13 and 14, persisting through middle and late adolescence, with stronger effects among early-maturing girls [33].

The purpose of the present study is to analyze gender differences in subjective wellbeing in Spanish children and adolescents by exploring three different components: life satisfaction, positive and negative affect and purpose in life.

## 2. Methods

### 2.1. Participants

Participants in the current study were recruited from the UP&DOWN study [41], a 3 year longitudinal study designed to assess the impact of physical activity and sedentary behaviors, lifestyle factors and health indicators over time, and to identify the psycho-environmental and genetic determinants of physical activity in Spanish children and adolescents. Participants were recruited from schools from the regions of Madrid and Cadiz, Spain. At a baseline, a sample of 2225 children and adolescents aged 6–20 was obtained. For the present study, the cross-sectional measure of the third year was analyzed because the assessment of evaluative wellbeing was only available in the last year. A total of 1407 youths aged 9–20 were considered, those with complete data on all subjective wellbeing indicators. Data were collected between September 2013 and June 2014. All variables employed in this study were collected by questionnaire and are self-reported. Before participating in the UP&DOWN Study, parents and school supervisors were informed by letter about the purpose of the study. Written consents were obtained for parents/guardians and participants. The UP&DOWN Study was approved by the Ethics Committee of the Hospital Puerta de Hierro in Madrid and the Bioethics Committee of the Spanish National Research Council.

### 2.2. Measurement Instruments

#### 2.2.1. Subjective Wellbeing Indicators

Three different subjective wellbeing indicators related to evaluative wellbeing, hedonic wellbeing and eudemonic wellbeing were assessed.

Evaluative wellbeing was measured using the subjective happiness scale (SHS). This is a four item scale in which two items ask respondents to rate themselves using both absolute rating and ratings relative to peers; the other two items offer brief descriptions of happy and unhappy individuals and ask respondents the extent to which each characterization describes them. The score ranges from 4 to 28, higher scores indicating better evaluative wellbeing. The scale showed substantial correlations (from 0.61 to 0.72) with traditional measures of evaluative wellbeing such as the satisfaction with life scale [42]. The Spanish version of the SHS, which observes the same structure as the original, was employed in this study. It evidences adequate internal consistency with a Cronbach’s alpha of 0.81, appropriate test-retest reliability and convergent validity for research purposes [43]. In the present sample, a Cronbach’s alpha of 0.70 was found for this scale.

Hedonic wellbeing was measured through the positive and negative affect schedule (PANAS). This instrument comprises of two scales of 10 items each, providing two independent measures of positive and negative affect, which are linked with positive feelings such as joy or pleasure and negative feelings such as anxiety and sadness. The score ranges from 10 to 50, with higher scores indicating higher positive and negative feelings respectively. All the items have a five point Likert scale that ranges from very slightly or not at all (1) to extremely (5) [44]. This scale is one of the most used measures of positive and negative affect [45]. In the present study the Spanish version for children and adolescents (PANASN) was used, which respects the same bidimensional structure and shows adequate internal consistency, convergent and discriminant validity [46]. In the present sample, a Cronbach’s alpha of 0.74 was found for this scale.

Finally, eudemonic wellbeing was assessed using the Children´s Hope Scale (CHS). This scale conceptualizes children´s hope in terms of positive expectancies, suggesting that children’s goals are related to two components: agency and pathways. Using a six-item dispositional self-reported index, the Children’s Hope Scale reflects the combination of agentic and pathways thinking toward goals, where the agency is reflected in initiating and sustaining actions and pathway thoughts are reflected in beliefs in one’s capacities [47]. The score ranges from 3 to 36, with higher scores indicating higher children’s hope. Hope (as measured with this scale) has shown discriminant properties for eudemonic wellbeing because of its involvement in goal-directed behaviors [8]. The Spanish version showed adequate internal consistency with a Cronbach’s alpha of 0.76 as well as adequate convergent validity; the original theoretical structure with two factors was respected [48]. In the present sample, a Cronbach’s alpha of 0.81 was found for this scale.

#### 2.2.2. Covariates

The covariates considered in the analysis were age and socioeconomic status (SES). To measure SES the family affluence scale (FAS) was employed. This is a 4-item scale corresponding to the questions: Does your family own a car, van or truck? Do you have your own bedroom for yourself? How many computers do your family own? During the past 12 months, how many times did you travel away on holiday with your family? Based on the scores obtained, adolescents were categorized in those with low, medium or high SES [49,50].

### 2.3. Data Analysis

The characteristics of the sample and main variables of interest such as age, gender, SES, evaluative wellbeing, positive affect, negative affect and hope were described as frequencies (percentages) or mean ± standard deviation (SD). The analyses were stratified by children and adolescents. Statistical differences between males and females in both groups were identified by the Chi squared test for categorical variables, and ANOVA tests were used for age and all subjective wellbeing variables.

In addition, several linear regression models were constructed in order to analyze the relationship between subjective wellbeing and gender. Linear regression has been implemented in various fields of study, particularly in research into health and social sciences [51,52]. Those variables of interest that could show significant effects were included as covariates in the models. Thus, linear regression models were adjusted by age and SES. The results of the models are presented as standardized coefficients (ß) with their statistical significance (*p*), with the boys as reference group, being the dummy coding of boys 0, and 1 for girls. The goodness of fit was assessed using R-squared (R^2^). The analyses were carried out with SPSS v.x for Windows (IBM, Armonk, New York, NY, USA), and the level of statistical significance was set at α < 0.05 (two-tailed).

## 3. Results

### 3.1. Descriptive Analysis

The study sample with valid measures in the third year comprised 1407 youths aged between 9 and 20 years, 722 males and 685 females. Given the wide range of ages, the sample was divided into children for those aged between 9 and 12 years (mean = 11.2) and adolescents for those aged between 13 and 20 years (mean = 15.3). Regarding the socioeconomic status (SES), 688 (50.4%) youths were identified as belonging to a high SES, 552 (40.5%) as a medium SES and 124 (9.1%) as low SES. Differences between groups were considered in the descriptive analysis (see Table 1).

As shown in Table 1, no significant differences were found in age between males and females, both in children (*p* = 0.455) and adolescents (*p* = 0.830). No SES differences were observed between males and females, in children (*p* = 0.092) or in adolescents (*p* = 0.618).

No significant differences were shown between male children and female children in any of the subjective wellbeing indicators (Table 1). Regarding adolescents, no significant differences were found in evaluative wellbeing between males and females (*p* = 0.391). The results changed when analyzing positive affect, where significant differences were found between males and females (*p* = 0.019): negative affect with higher scores in females than males *(p* < 0.001); and hope, with higher scores in males than females (*p* = 0.025).

Correlations between age and wellbeing indicators, in participants as a whole and by sex are shown in Table 2. As a whole, age of the participants was positively correlated with negative affect score (r = 0.141, *p* < 0.001). When the same analysis was carried out in male and female participants separately, significant differences were observed between age and evaluative wellbeing (subjective happiness scale; r = 0.080, *p* = 0.034) and negative affect score (r = −0.075, *p* = 0.048) in males, while age and negative affect score (r = 0.199, *p* < 0.001) were correlated in females (see Table 2).

Correlations between SES and wellbeing indicators in participants as a whole and by sex are also shown in Table 2. As a whole, significant differences were found between SES and evaluative wellbeing (*p* = 0.012), positive affect (*p* = 0.001) and hope score (*p* < 0.001). In males, associations were detected between SES and evaluative wellbeing (*p* = 0.004) and between SES and hope score (*p* < 0.001). In females, associations detected were between SES and positive affect (*p* = 0.005) and between SES and the hope score (*p* < 0.001; see Table 2).

### 3.2. Relations Between Evaluative Wellbeing and Gender

Linear regression models for the association between evaluative wellbeing and gender are presented in Table 3. No relevant variations were found between unadjusted and adjusted models and therefore only adjusted results will be described in this section. No significant relations were found between evaluative wellbeing and gender in the group of children (ß = 0.038; *p* = 0.370). Regarding the group of adolescents, no significant relations between subjective happiness and gender were found (ß = −0.030; *p* = 0.383; Table 3).

### 3.3. Relations Between Positive and Negative Affect and Gender

Linear regression models for the association between positive affect and gender are shown in Table 2. No significant relations between positive affect and gender were found in the group of children (ß = 0.023; *p* = 0.604). In the group of adolescents, a significant association between positive affect and gender was observed (ß = 0.084; *p* = 0.016).

Linear regression models for the association between negative affect and gender are shown in Table 2. No significant relations between negative affect and gender were found in the group of children (ß = 0.059; *p* = 0.173). Regarding the group of adolescents, linear regressions indicate significant relations between negative affect and gender (ß = 0.250; *p* < 0.001; Table 2).

### 3.4. Relations Between Hope and Gender

Linear regression models for the association between hope and gender are shown in Table 2. A significant relation between hope and gender was found only in adolescents. Male and female children did not show significant differences in hope (ß = −0.007; *p* = 0.871). However, significant differences were found between male and female adolescents in hope (ß = −0.079; *p* = 0.024).

## 4. Discussion

The aim of this study was to analyze gender differences in subjective wellbeing among Spanish children and adolescents using a three component model including life satisfaction, positive and negative affect and purpose in life. The results are in line with previous findings, highlighting the existence of gender differences in subjective wellbeing in adolescents [28,29,30,31,32,33,34]. These differences were found in hedonic and eudemonic wellbeing, but not in evaluative wellbeing. Moreover, differences were found among adolescents, but not among children.

### 4.1. Relationship Between Evaluative Wellbeing and Gender

Although some studies have found significant differences in life satisfaction between male and female children and adolescents [53,54,55], our findings did not show such differences. Our results were more congruent with those authors who have observed that life satisfaction is not related to gender condition [56]. Although no measures of masculinity and femininity were included in the study, specific measures of these constructs may have contributed to identify the expected gender differences in evaluative wellbeing, as previous studies have found [57].

### 4.2. Relationship Between Hedonic Wellbeing and Gender

Significant differences in the positive and negative affect between male and female adolescents were found. Female adolescents scored higher in both the positive and negative affect. Our results are supported by previous findings in which higher levels of negative affect in female adolescents were reported [58,59,60,61,62]. Although the literature is more limited in the case of positive affect, some authors have also suggested higher levels of positive emotions among girls [63] or even similar levels but with more intense positive emotions [24]. Possible explanations for this phenomenon entail biological, psychological and social factors.

Biological hypotheses suggest that mood changes associated with reproductive cycles [64], pubertal factors [60,65] and sex hormones [66] could be behind gender differences in the affect.

Psychological explanations offer a great variety of arguments in order to understand these differences. Some studies have found that male adolescents show a higher sense of efficacy in regulating their negative affect and a more robust self-esteem as well as greater levels of hedonic balance [63]. A greater exposure to stressors among girls has also been highlighted, such as body image concerns [67] and different coping strategies in response to stressful situations [68]. Cognitive factors have also been included as predictors of negative affect among females, suggesting a higher presence of negative life events and a greater cognitive vulnerability [59].

Finally, social factors should also be considered [60]. Social roles associated with femininity have been traditionally linked to concerns about interpersonal relations [68], girls being more emotion-focused and ruminative coping than boys [69] and positively linked to depressive states [29]. Moreover, increasing levels of masculinity during adolescence may improve mental health, suggesting that masculinity, but not femininity, is a central axis of advantages and disadvantages across adolescence in psychological wellbeing and psychological health [70].

### 4.3. Relationship Between Eudemonic Wellbeing and Gender

Higher levels of hope were observed among male adolescents. Literature about gender differences in hope in adolescence is scarce, although some studies have suggested that a robust hope is particularly important among girls, since the traditional feminine gender role is a disadvantage in overcoming traumatic life events [71]. Other findings have showed a general decline of hope over time both in male and female adolescents, but a higher decline among girls [72]. In addition, it has been stated that male and female adolescents show different patterns of hopes and fears (self-views), with girls rating significantly more fears (understanding fears as negative images of oneself in the future) than boys, which could negatively impact their self-esteem [73]. Possible explanations for this phenomenon may be found in traditional gender roles. In general, men can be described as more confident than women [74]. Women’s lower confidence is reflected in the fact that businesswomen generally report lower levels of profitability [75] and also in the fact that women judge themselves more harshly [76]. As a consequence, women are generally considered to be more risk averse [77]. All these aspects seem to be connected with constructs such as purpose in life, personal meaning and growth [78].

### 4.4. Gender Differences in Subjective Wellbeing: Children Versus Adolescents

Finally, the fact that gender differences in hedonic and eudemonic wellbeing were found in adolescents but not in children is aligned with previous research in this area suggesting that gender differences in wellbeing begins in early adolescence, between the ages of 12 and 15 years [54,62,64,79]. Possible explanations involve hormonal changes, stressful life events and less efficacious coping strategies that emerge among girls at this age [54,80].

## 5. Limitations and Future Research

The results of the present study should be interpreted taking into account some limitations. As in all cross-sectional research, it was not possible to establish causality. Future studies are needed that can infer causality from these associations. Future longitudinal research could be focused on the impact that gender has on subjective wellbeing, and the conditions that promote or inhibit this effect. In addition, our sample was not representative of the population of Spanish children and adolescents, which reduces the generalizability of our results. Finally, although age and SES were controlled in the multivariable analyses, other confounders should be explored and may have resulted in residual confounding.

## Figures and Tables

**Table 1 ijerph-17-02732-t001:** General characteristics of the study sample.

	Children			Adolescents		
	Male	Female	*p*	Male	Female	*p*
*n* = 1407	*n* = 297	*n* = 284		*n* = 425	*n* = 401	
Age (mean, SD)	11.2 (0.4)	11.2 (0.4)	0.455	15.3 (1.5)	15.3 (1.5)	0.830
Sex (%)	51.1	48.9		51.5	48.5	
SES						
Low (%)	29.9	38.3	0.092	61.4	62.1	0.618
Medium (%)	54.2	46.0		34.7	32.8	
High (%)	15.9	15.7		3.9	5.1	
SHS (mean, SD)	17.2 (4.1)	17.5 (4.2)	0.406	18.0 (3.4)	17.8 (3.5)	0.391
Positive Affect (mean, SD)	23.1 (3.7)	23.5 (3.4)	0.253	22.9 (3.1)	23.4 (3.1)	**0.019 ***
Negative Affect (mean, SD)	15.9 (3.5)	16.3 (3.3)	0.153	16.3 (3.4)	18.1 (3.4)	**<0.001 ***
Hope (mean, SD)	23.0 (5.5)	23.2 (5.6)	0.727	23.3 (5.0)	22.5 (4.8)	**0.025 ***

* Differences between boys and girls were analyzed by ANOVA and Pearson chi square for continuous and categorical variables, respectively. SES = socioeconomic status; SHS = subjective happiness scale.

**Table 2 ijerph-17-02732-t002:** Correlations between age/SES and wellbeing indicators, in participants as a whole and by gender.

**Age (r/p) ^a^**	**Males**	**Female**	**All**
*n*	722	685	1407
SHS	**0.080, 0.034**	−0.025, 0.511	0.038, 0.164
Positive Affect	−0.059, 0.118	−0.018, 0.648	−0.017, 0.539
Negative Affect	**−0.075, 0.048**	**0.199, <0.001**	**0.141, <0.001**
Hope	0.06, 0.884,	−0.048, 0.215	−0.020, 0.449
**SES ^b^**	**Males**	**Female**	**All**
*n*	722	685	1407
SHS	**0.004**	0.363	**0.012**
Positive Affect	**0.062**	**0.005**	**0.001**
Negative Affect	0.796	0.947	0.652
Hope	**<0.001**	**<0.001**	**<0.001**

^a^ Pearson correlation test, (Pearson correlation coefficient/*p*-value). ^b^ ANOVA test, *p*-value. SHS: subjective happiness scale.

**Table 3 ijerph-17-02732-t003:** Linear regression models for the association between subjective wellbeing and gender.

**Unadjusted Model**	**SHS**		**Positive Affect**		**Negative Affect**		**Hope**	
	ß	*p*	ß	*p*	ß	*p*	ß	*p*
Children	0.035	0.406	0.049	0.253	0.061	0.153	0.015	0.727
Adolescents	−0.030	0.391	0.083	**0.019**	0.249	**<0.001**	−0.079	**0.025**
**Adjusted Model**	**SHS**		**Positive Affect**		**Negative Affect**		**Hope**	
	ß	*p*	ß	*p*	ß	*p*	ß	*p*
Children	0.038	0.370	0.023	0.604	0.059	0.173	−0.007	0.871
Adolescents	−0.030	0.383	0.084	**0.016**	0.250	**<0.001**	−0.079	**0.024**

Adjusted model was adjusted by age and socioeconomic status (SES).

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
