# Peer review of "The Investigation of Gender Differences in Subjective Wellbeing in Children and Adolescents: The UP&DOWN Study"

_ijerph, 2020, doi:10.3390/ijerph17082732_

Round 1

Reviewer 1 Report

Review of IJERPH, “The Development of Gender Differences in Subjective Wellbeing in Children and Adolescents.  The UP & DOWN Study”

In general, the data has potential but, in my opinion, the analyses are incomplete.  The authors should report the correlation between age and the DVs to examine possible age effects.  Age is continuous and so are the DVs, so correlate both.  Report these correlations for all participants and then separately by sex.  At no point do the authors examine if the DVs or wellbeing, happiness, and/or hope differ between the SES groups.  These analyses should be completed before the regressions, which strip away the variance due to SES. For the regressions reported in Table 2, the analyses are illogical.  If age effects are removed, then what do the authors think the results are representing?  Conducting analyses within two age groups without age variance does not make any sense.  In addition, the authors report what other researchers have found with respect to the reliability of the scales, but do not report the values found for their sample.  This information is needed.

The other major concern I have is the statement made by the authors on page 2, line 73.  Studies which have looked at sex and/or gender and wellbeing are anything but “scarce”!  A quick Google Scholar search revealed 3,080,000 articles.  I would suggest deleting that claim altogether.

Less major comments:

- Abstract – if citing test titles, the titles need to be referenced

- Abstract lines 28-29, the sentence is incorrect as it suggests that an effect was found with children which was not what the authors claim.  I would suggest deleting this sentence

- Line 52, phrase, “it understands” is incorrect.  A concept or theory cannot “understand” anything.  Try using “define”

- Line 59 – define ESM

- Lines 70 and 79 – do not use the term “proven”.  Use a word such as “reported” or “demonstrated”

Author Response

Please see attacement.

Reviewer 2 Report

Due April 6, 2020

Review request: International Journal of Environmental Research and Public Health

Manuscript ID: ijerph-766805

Title:      The development of gender differences in subjective wellbeing in children and adolescents. The UP & DOWN Study

Synopsis

This subset of the UP&DOWN study (Cadiz and Madrid, Spain), composed of 1,407 children and adolescents who contributed data at the third year, was used for this cross sectional study. Data on gender (sex), age, socioeconomic status (SES), three domains of subjective wellbeing: evaluative, hedonic, and eudemonic measurements were used for chi-square, ANOVA, and linear regression analyses. Operational definitions of evaluative wellbeing were measured by the Subjective Happiness Scale.  Hedonic wellbeing was operationalized by the Positive and Negative Affect Schedule (PANAS), and eudemonic wellbeing was operationalized by the Children’s Hope Scale. Results demonstrated that the respective gender groups were comparable for age and SES. With linear regression assigning the boys as reference (dummy code = 0), statistically significant difference were detected for gender in hedonic positive (β = 0.084, p=0.016) and negative (β = 0.250, p=0.001) affect and eudemonic (hope) (β = -0.079, p=0.024) measurements, but not for evaluative wellbeing (β = -0.030, p=0.383). These findings are consistent with previously reported findings. Further investigation is called for to find the causality and developmental factors of this differential subjective wellbeing by gender.

Reviewer's conflict of interest: None

Comment to authors

Thank you for the opportunity to review this interesting article. My comments on statistical methods are not a mandatory revision request; however, it will be beneficial for the reader to understand the options for analysis and justification of the use of linear regression analysis for a discrete variable. Minor typos and comments are listed below:

    1. A minor issue for a computer search term; however, should “UP & DOWN” be “UP&DOWN”? Previous published articles used “UP&DOWN” without spaces. Please check the formal name of this study.
    2. Replace a period “.” after “adolescent” with a colon “:”.
    3. Is “Investigation” a more suitable term instead of “development”? “The investigation of gender differences in subjective wellbeing in children and adolescents: The UP&DOWN Study” This study cannot directly imply the findings in children versus adolescent age groups solely as a ‘development’ issue; perhaps additional social and personal developmental variables need to be added for future studies as explained in your limitations section. Thus, ‘development’ does not seem an appropriate term for the title.
  1. Abstract. Well written but please refer my synapsis as well
  2. Introduction. Two sections are well organized and connected.
  3. Methods:
    1. Line 70, could you elaborate on the difference between “gender differences” and “gender divergences.” I think I know what it is but would like some clarification.
    2. Line 92; 102; 103, “UP & DOWN”
    3. Line 153, “several linear regression models” were constructed in order to analyze the relationship between subjective wellbeing and gender.” Line 158 states: “the boys as reference group.” Is the dummy coding of boys to be 0, and girls to be 1? Please indicate the assigned dummy codes. I assume authors used the regression model to imply the ‘relationship’; however, for this cross sectional data from 3rd year participants in the UP&DOWN study, the difference in subjective wellbeing should be detected by multiple ANCOVA (MANCOVA). MANCOVA can control variables such as SES and age and the interaction between types of subjective wellbeing can be analyzed without type 1 error adjustment. This comment is not mandatory because I do not know the characteristics of distribution of the 1,407 samples. The simple linear regression equations with dummy coding is fine, but please discuss the limitations and assumptions of your method.
    4. Line 160 – 161, contains duplicated information. The significance level is usually referred to as the α
  4. Results:
    1. Line 164, “the third wave.” Line 92, “a 3-year longitudinal study,” Line 98, “only available in last stage.” I recommend using consistent expressions for the 1,407 subjects.
    2. Table 2 (line 188), this is format issue so it should be corrected for publication. Please line up the columns.
    3. Line 189, “Boys are considered as the reference group,” meaning the beta reflects the relative value for girls. For instance, the Hope has a negative beta, meaning the girls’ hope score is lower than boys’. The girls’ negative affect is higher than boys’. I just would like to make sure what I am reading is correct.
  5. Discussion. Well sectioned and organized.

End of review.

Round 2

Reviewer 1 Report

Although I still wonder about the logic of categorizing/dichotomizing children from adolescents almost arbitrarily, I am satisfied that my concerns were addressed and that the edits to the text were conducted.